# Hyperprogression Under Immune Checkpoint-Based Immunotherapy—Current Understanding, The Role of PD-1/PD-L1 Tumour-Intrinsic Signalling, Future Directions and a Potential Large Animal Model

**DOI:** 10.3390/cancers12040804

**Published:** 2020-03-27

**Authors:** Mikolaj Kocikowski, Katarzyna Dziubek, Maciej Parys

**Affiliations:** 1International Centre for Cancer Vaccine Science, University of Gdansk, Wita Stwosza 63, 80-308 Gdansk, Poland; mikolaj.kocikowski@ed.ac.uk (M.K.); katarzyna.pietrzak@phdstud.ug.edu.pl (K.D.); 2The Royal (Dick) School of Veterinary Studies and The Roslin Institute, University of Edinburgh, Easter Bush Campus, Midlothian EH25 9RG, UK

**Keywords:** hyperprogression, hyperprogressive disease, tumour-intrinsic signalling, cancer, immunotherapy, comparative oncology, canine model, immune checkpoint blockade, PD-1, PD-L1

## Abstract

Immune evasion is a major challenge for the development of successful cancer treatments. One of the known mechanisms is the expression of immune checkpoints (ICs)—proteins regulating the immune cells activation. The advent of immunotherapy using monoclonal antibodies (mAbs) to block the immune checkpoint receptor-ligand interaction brought about a landslide improvement in the treatment responses, leading to a prompt approval of such therapeutics. In recent years, it was discovered that a subset of patients receiving IC blockade treatment experienced a previously unknown pattern of treatment response called hyperprogression (HP), characterised by rapid deterioration on initialisation of the therapy. HP represents an urgent issue for clinicians and drug developers, while posing questions about the adequacy of the current clinical trial process. Here, we briefly summarise the state of knowledge and propose new directions for research into HP mechanisms, focusing on tumour-intrinsic signalling of IC proteins malignantly expressed by cancer. We also discuss the potential role of spontaneously occurring canine cancer in the assessment of immunotherapeutics, which can provide the missing link between murine and human studies.

## 1. Introduction

Cancer is an urgent problem facing the biomedical field. Its hallmark ability to modulate the host immune system and evade destruction represents a major challenge for the development of successful treatments. One of the key discovered mechanisms of immune evasion is based on the expression of proteins belonging to the immune checkpoint (IC) group. These ligands interact with receptors of the host immune cells to regulate their activation state. The increasingly common use of immunotherapy in cancer treatment, particularly the implementation of the IC blockade (ICB), preventing previously mentioned interaction, has proven a breakthrough treatment in some cancer types. While not all patients respond to this line of therapy, a substantial subset experiences rapid disease progression—a recently described phenomenon called hyperprogression (HP) or Hyperprogressive Disease (HPD). While the clinical data and some biological explanations have been comprehensively described before [1,2,3,4,5,6,7,8,9,10,11,12,13,14,15,16,17], this review aims to discuss several unexplored questions and mechanisms that may contribute to HP, with a particular focus on tumour-intrinsic PD-1/PD-L1 signalling. Importantly, we point out the limitations of the studies in the murine model and discuss the spontaneously occurring canine cancer as a better alternative for preclinical trials. Dog model is capable of closely resembling the characteristics of human cancer-immune system synapse and could serve as a strategy for gaining early insight into adverse effects. Additionally, this approach has a potential to reduce the bench-to-bedside distance by enabling shorter clinical trials. The improved efficiency of drug discovery pipelines would benefit all stakeholders.

### Immunotherapy using Immune Checkpoint Blockade

Immunotherapy is a treatment modulating the activity of the host immune system. The ratio of improved survival to the extent of adverse effects is favourable for immunotherapy when compared to classic cancer therapies [18,19]. The most prevalent immunotherapy approach uses recombinant monoclonal antibodies (mAbs). Since the FDA approval of the first cancer-targeting mAb Rituximab, at least 35 more have been introduced to the clinical practice [20]. Immune checkpoints are proteins that modulate cellular responses to immunogenic stimuli, leading to either inhibition or activation of immune cells. In a healthy organism they are essential for maintaining self-tolerance. There are multiple known activatory and stimulatory ICs. Currently, there are two inhibitory ICs in the clinical spotlight, Programmed Cell Death Protein 1 (PD-1) and Cytotoxic T-cell Antigen 4 (CTLA-4) receptors together with their ligands: PD-L1, PD-L2 [19], and CD80, CD86, respectively. PD-1 is expressed mainly on T-lymphocytes and NK cells [21], and its most studied ligand—PD-L1—in a variety of healthy tissues, especially after cytokine stimulus, as well as on antigen presenting cells (APCs) [22,23]. PD-L1 is also expressed by the cells of multiple cancer types [19]. It binds the PD-1 receptors of nearby T-cells, preventing them from attacking the tumour. Monoclonal antibodies against the IC receptors and ligands were developed to block their interaction and prevent the resulting T-cell energy (Table 1). This approach is known as Immune Checkpoint Blockade (ICB; or ICI for “inhibition”). The increasingly common use of ICB immunotherapy against PD-1 and CTLA-4 induced remarkably long-term responses in patients with multiple cancer types, particularly malignant melanoma [24,25,26]. The impact of this therapy on human oncology was highlighted in 2018, when the Nobel Prize was awarded to James P. Allison and Tasuku Honjo for its discovery.

## 2. Atypical Responses to ICB

Owing to a different mechanism of action as compared to the conventional chemotherapy or targeted therapies, ICB immunotherapy frequently induces atypical response patterns such as long-term remissions observed in multiple cancers. Melanoma is a prime example, with close to 50% of patients experiencing durable response. Such clinical trial results allowed for a breakthrough status and fast-track introduction of this immunotherapeutics class. The speed of approval did not however lend itself to a careful interpretation of the treatment responses in the non-responding groups. Two other patterns have been described that were not commonly observed in the past: pseudoprogression and hyperprogression.

### 2.1. Pseudoprogression

Upon initialisation of ICB, in several patients the disease progressed rapidly as measured by volumetric or 2-dimensional assessment of solid tumour size. Subsequently, the tumour volume decreased, leading to a successful treatment response. This phenomenon is considered to stem from inflammation and accumulation of tumour-infiltrating lymphocytes (TILs) that increase the tumour mass temporarily. Hodi et al. defined pseudoprogression as a tumour burden increase by at least 25% that turns out to not represent progressive disease during the following assessment. Interestingly, pseudoprogression can appear either within the first 12 weeks of treatment or in a more delayed form [27]. It became crucial to distinguish pseudoprogression—affecting less than 10% of patients [3,27,28]—from real progression. In the initial trials pseudoprogression frequently led to unnecessary discontinuation of the beneficial treatment. Since such an aberrant pattern of response was not previously recognised for chemotherapeutic treatments, the commonly used treatment response evaluation criteria RECIST were proven inadequate and new sets of guidelines were developed: irRC (immune-related response criteria) [28], irRECIST (immune-related RECIST) [29], and iRECIST (immune RECIST) [30]. The current recommendation, in the updated RECIST criteria, is to recheck the patients at least 4 weeks after the diagnosis of potential progressive disease to ensure that the observed pattern is not pseudoprogression.

### 2.2. Hyperprogression

The concept of hyperprogression (HP) or Hyperprogressive Disease (HPD) was introduced by Champiat et al. in 2016, based on the observation that a subset of patients receiving ICB experiences extremely rapid disease progression that leads to fast patient deterioration [11]. Importantly, in these patients, the survival time is often shorter than 2 months [31]. Attempts by multiple groups to capture the cases of HPD led to several definitions of the phenomenon (summarised in Table 2) and were comprehensively described by Kim et al. [31]. A lack of a unified definition adds to the controversy around HP and makes it difficult to compare studies or to combine their results for higher statistical power. The diagnosis is based on quantifying different disease progression markers at different timepoints from the treatment initialisation or as compared to a reference period before treatment. While seldom available, pre-therapy tumour growth kinetics data are needed to establish the individual baseline. Depending on the study and elected definition, HP affected 5%–37% of cancer patients. The real incidence may be higher, considering that patients who deteriorated fastest could not be fully evaluated. HP was observed in the following types of cancer: non-small-cell lung carcinoma (8%–37%), melanoma (6%–34%), gastrointestinal (15%–21%), head and neck (9–18% and 29% in case of Squamous Cell Carcinoma), gynecological (16%), other lung (10–15%), cutaneous squamous cell carcinoma (9%), renal (5%–7%), colorectal (6%), urothelial (6%) [4,11,32,33,34,35,36,37,38]. All the discussed studies pertained to PD-1/PD-L1-based immune checkpoint blockade.

## 3. Diagnosing Hyperprogression

It is essential to detect and distinguish between progression, pseudoprogression and hyperprogression early. Since in most cases the tumour growth rate (TGR) data from before the immunotherapy is not available, the distinction is not possible by basic imaging of changes in the tumour size. Berge et al. demonstrated that in the presence of such data, TGR is a clinically relevant predictor of overall survival [40]. Attempts to track disease development by regular genetic testing of tumour biopsies were inconsistent, troublesome for the patient and often contraindicated [41,42]. Schutz et al. applied a liquid biopsy approach with promising results. They assessed cell-free circulating DNA from the serum [41] by quantitative analysis of chromosomal instability, avoiding the limitations of methods focusing on specific genes. The method was versatile, minimally invasive and able to distinguish real from illusive progression with 90% accuracy when radiological evidence uniformly suggested disease progression. It also appeared capable of distinguishing HP from pseudoprogression. Jensen et al. proposed a similar approach [43]. Another liquid biopsy method was reported by Zuazo-Ibarra et al. [44], who quantified the amount of circulating senescent CD4+ T-cells (Tsens) before ICB treatment and could reportedly stratify NSCLC patients into responders and intrinsic non-responders (including hyperprogressors) with 100% specificity and 75% sensitivity. High levels of Tsens before immunotherapy indicated responders and decrease in Tsens numbers after the first treatment cycle predicted good response. The decline was putatively a result of G1 phase exit or recruitment from blood to the tumour site. Conversely, a proliferative increase in Tsens cells numbers indicated hyperprogression. The authors concluded that extensive validation would be necessary to apply the findings clinically in NSCLC and wider. Boeri et al. found potential prognostic value in profiling the immune environment by analysis of microRNA from plasma [45]. Overall, liquid biopsy-based techniques hold promise for a reliable, patient-friendly stratification and disease tracking method applicable in the clinic or perhaps in real time in the future. A combination of several approaches may provide the most accurate assessment.

## 4. Markers Associated with HP

Associations were made with increased age [11], higher lactate dehydrogenase (LDH) concentration in the serum [31], female sex [35], previous irradiation of the tumour area [37], pre-existence of liver-located or more than two metastatic sites [46], MDM2/MDM4 and EGFR genetic alterations [32,47] and mutations in a variety of oncogenes [48]. Most results were not replicated by other studies [49]. In a recent meta-analysis published in ‘*Cancers*’, Kim and colleagues found that only five factors can be considered statistically significant in the general patients’ pool. HP was associated with elevated serum LDH concentration, presence of more than two metastatic sites, liver metastases and Royal Marsden Hospital (RMH) prognostic score equal 2 or more. Strong expression of PD-L1 was inversely associated with HP [31]. The authors called for efforts to formulate a standardised definition since the lack thereof made comparisons of the available studies difficult. The main limitations in most studies were lack of satisfactory pre-treatment data and small cohorts. These prevent reliable conclusions for specific patient groups (by cancer type, stage, histology, previous treatments or lack thereof and more), hence the associations lacking statistical significance in small studies may turn out important in specific subpopulations. If the MDM2/4 amplifications and EGFR mutations or other activatory mutations would prove to be a part of the HP mechanism, co-administration of ICB with inhibitors of these proteins could potentially prevent HP. It is hypothesised that the mechanistic effect of MDM2/4 amplifications is alternative to the classical p53-regulatory role and may be based on a co-amplification of the actual driver gene [50].

## 5. Postulated Mechanisms

Multiple mechanisms have been proposed to be involved in development of hyperprogression (Figure 1).

Kamada et al. described a rapid expansion of FoxP3 T-regulatory (T-reg) cells in gastric tumour patients with HP [51]. FoxP3 is a classical marker for T-regs, which are responsible for inducing immune tolerance. Importantly, the expansion of those cells was associated with PD-1 expression by the effector fraction of T-reg cells (CD45RA-CD25highFoxP3highCD4^+^), which expanded upon PD-1/PD-L1 blockade, resulting in a strongly immunosuppressive microenvironment, allowing for rapid cancer cell proliferation. The clinical results were confirmed in vitro and validated in mouse models, strongly supporting the role of T-reg cells in hyperprogression. Disabling PD-1 with Cre-Lox system enabled authors to show that PD-1-deficient T-regs exhibited enhanced proliferation and immunosuppressive mechanisms. The results were analogous when PD-1 was blocked with a mAb. On the other hand, Nair et al. described inhibition of peripheral FoxP3 T-reg cells differentiation and a FoxP3 down-regulation through mTOR pathway under pembrolizumab-mediated PD-1 blockade [52]. The treated T-cells were less suppressive, but the study has not been validated in vivo. Curiously, Duruisseaux et al. established an epigenetic signature specifically predictive of response to PD-1-targeting ICB, which included FoxP1 methylation status. Unmethylated status was a predictor of survival. Strikingly, FoxP1 modulates the activity of FoxP3 [53,54].

Similarly, as previously hypothesised by Rauch et al. in their study on adult T-cell leukaemia/lymphoma under Nivolumab treatment [55], Koyama et al. observed a compensatory up-regulation of additional checkpoints under PD-1-targeting mAb treatment in immunocompetent murine models of lung adenocarcinoma and in two patients as well [56]. They observed that the anti-PD-1 mAbs were still bound to their target on T-cell surface at the time of progression, and no link was observed between progression and myeloid cells composition in the TME. However, in progressive cases, the CD4^+^ and CD8^+^ cells overexpressed another inhibitory IC receptor TIM-3 (T-cell immunoglobulin and mucin-domain containing-3). TIM-3 was predominantly detected on ICB mAb-bound, intratumoral T-cells. The abundance of TIM-3+ cells was proportional to the length of the ICB treatment. Increase in TIM-3^+^ cells was not observed in cases which responded to the treatment. The up-regulation of TIM-3 is one of the mechanisms of resistance to PD-1-based therapies [57]. There are several ongoing clinical trials which combine PD-1 and TIM-3 blockade. The results of these studies will give further insights into TIM-3 compensatory mechanisms and their potential role in the development of hyperprogression. There is an increasing interest in combining different ICB agents or administrating ICB together with other classes of therapeutics. Several treatment regimens are currently in trials, some with promising results. For instance, there is growing evidence for enhanced therapeutic efficacy of concurrent PD-1 and CTLA-4 blockade, and the combination of Nivolumab with Ipilimumab was recently granted an accelerated approval by FDA based on results from CheckMate-040 (for patients with hepatocellular carcinoma unsuccessfully treated with sorafenib). However, there are no reports yet on the rates of hyperprogression in combination therapies. Interestingly, in CheckMate-032, where metastatic esophagogastric cancer patients were treated with Nivolumab or combination of Nivolumab and Ipilimumab, the number of patients with increased tumour growth rate in the initial treatment stages was higher in the combination groups when compared to Nivolumab only [58]. This example further shows the need for inclusion of hyperprogression criteria in the evaluation of clinical trials studying ICB agents alone or in combination.

Stein et al. demonstrated that CD8^+^ T-cells that become activated and interact with breast cancer cells but are incapable of lysing them elicit dedifferentiation cascade in the target tumour cells, inducing cancer stem cell (CSC) formation [59]. They hypothesised such an effect could stem from ICB treatment, but the biological mechanism remains to be discovered.

Kudo-Saito et al. observed that in the murine model and in cell culture studies chemo- and immunotherapy activated putative cancer stem cells (CSCs) leading to aggressive proliferation and resistance. They suggested that HP may depend on the proportion of cancer cells elimination and dormant cells activation [60]. While the study focused on treatment-induced metastatic lesion growth, this observation could prove universal. It remains to be tested whether cells with CSC potential express ICs and perhaps can be a subject of tumour-intrinsic signalling. Okeya et al. observed a case of advanced lung adenocarcinoma transforming into small-cell carcinoma coinciding with HP and metastases after five weeks of pembrolizumab treatment in a 66-year old male smoker [61]. This effect could be a result of cell transformation. Alternatively, treatment could select for previously undetected pre-existing cells of the second cancer type and stimulate them, which could contribute to HP.

In 2017, Arlauckas et al. performed an in vivo imaging experiment by tracking fluorescently labelled anti-PD-1 mAbs, MC38 tumour cells, and tumour-associated macrophages (TAMs) injected into mice and following the interactions between those components [62]. They observed mAb accumulation at the PD-1+ target cancer cells and subsequent capture by PD-1- TAMs. The elucidated mechanism was mAb binding by specific Fc-gamma receptors, and the effect was abrogated in vitro by mAb-based FcR pre-blocking. The setup was adapted for testing a human mAb Nivolumab, and the results were replicated. The FcR pre-blocking was repeated in vivo which eliminated occurrence of non-responder mice, putatively through increased PD-1 exposure to available mAb. The effect was dependent on the Fc of the antibody and the FcRs expressed by the macrophage. The finding revealed fundamental considerations for immunotherapeutics design. In another study, Zhang et al. designed two anti-PD-1 mAbs of the same specificity but differing in the Fc sequence [63]. One did not bind FcRs and did halt the tumour growth. The other one was binding FcRs, which led to cross-linking macrophages with PD-1+ T-cells and phagocytosis of the latter ones. FcR interaction abrogated the anti-tumour effects and modified signaling and activities of both cell types. The authors pointed out that the affected Fc-gammaRI induces immune tolerance by modulation of inflammatory cytokines. Fc-gammaRI plays a role in the generation of M2 macrophages that are known to be tumour-supportive [64]. Finally, Lo Russo et al. provided evidence for macrophage reprogramming from M1 to M2 phenotype and suggested a detrimental role of Fc-gammaRIIb in human anti-PD-1 immunotherapy leading to HP-like effects [33]. The team demonstrated that Nivolumab-based Fab construct lacking the Fc portion did not elicit HP-like results in the experimental model. In genetically predisposed individuals, anti-PD-1 therapeutics with specific Fc sequences are likely to promote reprogramming of macrophages into an aggressive phenotype and result in HP. This could be solved by modifications to the current antibody therapeutics, aiming to disable FcR interaction. This will be further enabled by the currently shifting patent landscape regarding IgG heavy chain engineering [65].

### 5.1. Tumour-Intrinsic Signalling

Until recently, the mechanism of action of PD-1/PD-L1 blocking antibodies was considered to base solely on their ability to block the interaction between PD-1 receptor localised on the surface of T-cells and PD-L1 ligand expressed by cancer cells. In fact, success rates of PD-1/PD-L1 therapy vary between 20% and 90%, depending on the tumour type [66,67,68]. The remarkable outcomes were thought to result exclusively from the enhanced immune response. On the other hand, it has not been determined yet why some of the patients are refractory to ICB treatment or even demonstrate a rapid relapse. Since PD-1 and PD-L1 can be expressed by cancer cells, there may be another mechanism at play - PD-1/PD-L1 tumour-intrinsic signalling.

### 5.2. PD-1 Intrinsic Signalling

Multiple recent publications reported that cancer cells express not only PD-L1 but also PD-1 in tumour types such as melanoma, hepatocellular carcinoma, ovarian cancer, and NSCLC [69,70,71]. Until now, it had been believed that the PD-1 receptor could be expressed solely on haematopoietic cells [72]. Strikingly, in their case report Du et al. described rapid NSCLC progression (hyperprogression) in a patient treated with pembrolizumab [71]. Analysis of the tumour biopsy found NSCLC cells expressing the PD-1 receptor. A further study observed proliferation of tumour cells in the murine model after anti-PD-1 treatment. The molecular mechanism behind this phenomenon remains unknown, but authors detected an increase in cells expressing proliferation marker Ki67 and a decrease in apoptosis marker Caspase-3 [71]. In certain cancers, acute T-cell lymphocytic leukaemia as a prime example, PD-1 is proposed to function as a tumour suppressor [55]. The rapid growth of PD-1 expressing cancer cells after blocking PD-1 suggests its inhibitory role so widely observed in T-cells [73]. Nevertheless, despite well-described PD-1 inhibitory action in T-cells, PD-1 tumour-intrinsic signalling was reported to stimulate tumour growth in mice lacking adaptive immunity. Kleffel et al. observed increased growth of PD-1 expressing subpopulations of cancer cells, which was abrogated by PD-1 antibody administration. Additional analysis revealed selective stimulation of P-S6RP and P-eIF4E proteins indicating mTOR pathway activation regardless of PI3K/Akt signalling [69,74]. Protein tyrosine kinases (PTKs) may serve as a possible explanation of the different roles that PD-1 can play in cancer cells. PTKs widely interact with receptor tyrosine kinases (RTKs) modulating their downstream signalling that mostly transduces growth factors signalling. The outcome depends on a dual role of protein tyrosine phosphatases (PTPs) capable of suppressive or mitogenic activity. SHP1 and SHP2 are an example. In T-cells, SHP1 and SHP2 are recruited to PD-1 upon PD-L1 engagement to transduce inhibitory signalling. While SHP1 acts as a solely suppressive factor, mutations in SHP2 were reported in melanoma that evoke mitogenic function via the Akt or Ras pathways [75,76,77]. Appendix A summarises these mechanisms and points out the most important differences between PD-1 intrinsic signalling in T-cells versus cancer cells. Undoubtedly, a deeper understanding of PD-1 intrinsic signalling is necessary to conclude whether it is the mechanism of hyperprogression. The identification of factors involved in PD-1 intrinsic signalling may lead to changes in patient selection for PD-1 oriented immunotherapy.

### 5.3. PD-L1 Intrinsic Signalling

PD-L1 expression by cancer cells has been studied extensively and was associated with poor prognosis and metastatic disease [78]. Interestingly, PD-L1 was shown to interfere with major pathways in tumour cells independently of PD-1 interaction [79]. Clark et al. (2017) used mice melanoma and ovarian cancer cell models to demonstrate that PD-L1 intrinsic signalling increases tumour cells proliferation, possibly through mTOR signalling, as hinted by an elevated level of mTOR substrate P-70S6KT389. Strikingly, PD-L1 blockade suppressed the tumour growth in mouse xenografts [72]. This challenges our knowledge regarding mechanisms of the ICB therapy. Gupta et al. demonstrated that tumour PD-L1 increases tumour-initiating cells (TIC) generation, ultimately contributing to rapid tumour growth. Although this data comes from the murine ovarian carcinoma and melanoma cell lines, similar results were observed in a human ovarian cancer cell line. Consistency of observations across different cancer cell lines indicates that PD-L1 intrinsic signalling may be a universal phenomenon. Again, TICs were demonstrated to act through mTORC1 signalling, which was reduced by an mTOR inhibitor—rapamycin [72,80,81].

PD-L1 tumour-intrinsic signalling was also studied in glioblastoma (GBM). mRNA sequencing of PD-L1^high^ cells revealed activation of genes responsible for cell migration and motility. It was proposed that PD-L1 binds to Ras, then triggers the Ras/Raf/MAP/Erk cascade that regulates endothelial to mesenchymal transition, widely associated with enhanced tumorigenesis of GBM [82]. Besides effects on cancer intrinsic pathways, tumour PD-L1 expression was shown to protect tumour cells from IFN toxicity, which is a mechanism of antitumour immunity. The core effects of IFN activity are cell cycle arrest, cell senescence and induction of apoptosis [83]. While the works cited above provide examples of PD-L1 cancer intrinsic signalling, multiple contradictory studies were published as well. Lin et al. and Tang et al. pointed out the essential role of host adaptive immunity in response to both PD-1 and PD-L1 blockade, consequently questioning the importance of cancer PD-1/PD-L1 intrinsic signalling in this process [84,85]. There are contradictory reports regarding the role of tumoural PD-L1 expression in hyperprogression. Only one report found a significant inverse correlation between PD-L1 expression and HP in non-small-cell lung cancer [33]. The exact cause of this invert correlation is unknown, but two molecular mechanisms should be explored. In NSCLC, EGFR mutations and MDM2/4 amplifications are commonly found, and HP is a frequently recurring issue. Epidermal growth factor (EGF) signalling was shown to induce post-translational modifications of PD-L1 [86]. Changes to glycosylation patterns may affect both the receptor-ligand interaction and antibody binding, putatively leading to decreased tumour cell IHC staining, a decreased therapeutic efficacy and hyperprogression. Interestingly, MDM2 expression regulates VEGF expression in multiple cancers, including breast cancer and neuroblastoma [87]. VEGF was shown to regulate both PD-1 and PD-L1 [88]. In result, the increased expression of VEGF secondary to MDM2 amplification can putatively lead to aberrant ICB responses through changes in expression levels of both PD-1 and PD-L1 and changes in tumour-intrinsic signalling. Even if PD-L1 intrinsic signalling does not directly contribute to HP, a better understanding of PD-L1 signalosome in cancer cells may be vital for maximising the benefit of cancer immunotherapy. It seems that the network of both PD-1 and PD-L1 intrinsic signalling remains to be fully unravelled.

### 5.4. Other Unexplored Mechanisms

As mentioned earlier, Kim et al. note that the elevated LDH serum level significantly associated with HP has well established links with several mechanisms of tumorigenesis and immune evasion. High LDH indicates intratumoral hypoxia as well as extracellular environment acidification. The authors concluded that the link to HP is unknown [31]. It is known, however, that the solvent can influence the antibody functionality and the conformation of protein antigens [89]. Additionally, acidic tumour environment has been shown to reduce ICB efficacy [90]. Putatively, the change in the intra-tumoural environment parameters, as indicated by increased serum LDH levels and represented by pH reduction, may modify the specificity and affinity of the biotherapeutics. Djoumerska-Alexieva and colleagues demonstrated poly-reactivity and binding of IFN-gamma as a non-targeted control by therapeutic immunoglobulins as a result of low-pH exposure [89]. IgGs were affected to a different extent, depending on their manufacturing process. Taking this into consideration complicates drug design, testing, and production process, but also enables the development of innovative and more specific therapeutics [91,92,93]. Since Refae et al. demonstrated an association of HP risk with specific variants of PD-1, PD-L1 and VEGFR2 polymorphisms [94], it could be interesting to test how these variants affect the conformation of these proteins in different solvents. Additionally, the role of modifications in glycosylation patterns and other post-translational modifications (PTMs) of PD-1 and PD-L1 proteins is relatively unexplored. Glycosylation of PD-L1 is modulated by epidermal growth factor (EGF) signalling [95]. Moreover, EGF-induced glycosylation stabilises PD-L1 on the cell surface [86]. The effects of PTMs on the interaction between PD-1, PD-L1 and IC antibodies require further research.

The knowledge of HP incidence in different patient populations is lacking due to small cohorts and no data was found on HPD incidence between human races due to mostly Caucasian cohorts [96]. Califano et al. emphasise that due to all trial exclusion factors, the tested population differs significantly from the real patients [97]. For instance, the average patients’ age was 10 years lower in clinical trials participants, and those with chronic infections, comorbid disorders or pre-existing autoimmune disorders are excluded from clinical trials. Additionally, ICB can exacerbate, reveal, or cause autoimmune disorders [98]. The endocrine system is strongly linked to the immune system [99]. ICB treatments specifically can damage the endocrine system, affecting the thyroid, pituitary gland, and adrenal cortex among other organs [100]. If the reports linking patient’s age and sex with HP risk become confirmed, it could be partially explained by age and/or sex-dependent differences in the hormonal profile.

Binding of PD-1 by an ICB antibody blocks the PD-L1 binding site; however, it is known that the second ligand PD-L2 uses a different binding site and mechanism [101]. While they compete for the PD-1 receptor and possess similar affinity, they seem to elicit different responses and the role of PD-L2 is not explored to the same extent as PD-L1. There are some claims that both ligands may have counterintuitive functions and currently unknown receptors [102]. It could be possible for ICB to affect the interaction between PD-1 and its two ligands differently. Little is known about the impact of ICB on the stability of pre-existing PD-1-ligand complexes. The roles of soluble PD-1 and PD-L1 forms are being researched only outside of the HP context, but exosomal PD-L1 is a negative prognostic factor in melanoma [103]. Melanoma has been shown to recruit mesenchymal stem cells (MSC), induce PD-1 expression and transform them into melanoma-like pro-tumorigenic phenotype through exosomal signalling [104]. Neither the place of HP in this network nor the impact of intense ICB treatment on exosome production has been sufficiently assessed.

The microbiome is a well-known driver of inflammation and immune response. There is a body of evidence that gut microbiota impacts the outcomes of ICB treatment in melanoma patients [105,106,107,108,109,110], leading to increasing interest in therapeutic interventions into individual microbiome composition [105,109]. Furthermore, microbiota located at the site of the lesion seems to affect the history of the disease in resected pancreatic adenocarcinoma [111]. The individual microbiome appears to be of special importance for the outcomes of cancer therapy and in other diseases. To date, no studies have analysed microbiome data in the context of HPD.

Reports on the differences in the rate of HP between different ICB antibodies are contradictory [11,49]. Such comparisons performed within the same indication would be informative, but difficult without larger cohorts. Importantly, HP was described for immune checkpoint-blocking mAbs, but not other immunotherapeutics. A retrospective analysis could partially answer whether HP has been previously overseen or is it characteristic for ICB. The results could be then analysed in relation to differences in the Fc region sequences of the therapeutic antibodies, considering the putative FcR-mediated macrophage reprogramming pathway. If the Fc sequence differences result in different HP rate, strategic Fc engineering could help prevent TAM generation, avoid HP, and possibly exert other treatment-supportive functions [112] by harnessing the reprogramming effect to modify the behaviour of cells present in the tumour microenvironment (TME) for treatment benefit [113]. Furthermore, just as Saâda et.al found HP-associated polymorphic variants of IC proteins [94], it would be worthwhile to assess the heterogeneity of the Fc-receptors sequences between patients in the context of HP likelihood. Champiat et al. point out that T-cell behaviour in TME under ICB can be affected by mutations affecting IFN-γ signalling pathway, particularly JAK1/2 [10]. Precise profiling of T-cells of HP-affected patients could provide further insights. Sharon draws a possible analogy between HP and the significance of the PD-1/PD-L1 axis for the immune response to bacterial and viral infections [114], particularly the impact of ICB in a murine model of tuberculosis, where ICB leads to rapid and fatal disease exacerbation related to IFN-γ release by CD4^+^ T-cells.

Any mechanism by which ICB could help cancer cells decrease neoantigen expression and presentation, up-regulate alternative IC proteins or the production of immunosuppressive enzymes and cytokines, can be an important part of HP genesis. Finally, in such a complex disease, there may be no single mechanism or marker of HP applicable to all the cancer types and patient populations.

## 6. Arguments against HP

The crossing of Kaplan–Meier survival curves of the ICB arm and chemotherapy control arm in the early treatment period during ICB clinical trials such as Checkmate-057 [115] has been interpreted by some as evidence for HP existence. However, the observation that the disease was better controlled by chemotherapy in the initial 3–6 months could be explained by a delayed onset of ICB therapeutic effects. This would be a possible response pattern resulting from the unique mechanism of action of ICB-based immunotherapy. Some argue that the pattern interpreted as HP is just a natural development of disease, the rate of which is different between patients and not linear in time for an individual case. This indeed was difficult to prove wrong, as in most trials there was no reference data available on tumour growth rate (TGR) before ICB initialisation. However, some clinicians reported that looking at individual patients’ cases tells a different story than the average result, and some tumours presented progression at the rate never seen before. Champiat et al. performed a study where TGR was also assessed before the treatment [11]. They found that a subset of patients does indeed experience an unmistakably rapid increase of TGR upon ICB. Some opinions noted that rapid disease exacerbation upon treatment was observed with small molecule inhibitor drugs, and so is not exclusive to the ICB [5]. Ferrara and colleagues demonstrated that while HP was detected under chemotherapy, the incidence increased approximately 3-fold for ICB [39]. It appears that the existence of HP in a subset of cancer patients treated with ICB is no longer questionable, but its incidence under different treatments in specific patient subpopulations remains poorly understood just like the mechanisms ruling the pattern.

## 7. Proposed Strategies for Long-Term Problem Mitigation: Changing the Animal Disease Model

There is no denying that the murine model has brought insight into human cancer immunology and a wide spectrum of human diseases. However, only a fraction of therapeutics effective in mice enter human trials and of those, approximately 8% will pass the I phase of clinical trials [116,117]. The number reaching the clinics long-term could, in fact, become even lower if adverse effects of the HP scale continue to be identified at late development stages.

### 7.1. Drawbacks of the Murine Model

As a matter of fact, the murine model bears several bottlenecks that may contribute to failures in resembling human diseases accurately [118]. Laboratory mice are isolated from the external environment, kept in specific antigen-free conditions and frequently immunocompromised as patient-derived xenografts are a common adaptation of the murine model. Laboratory mouse strains are characterised by very low genetic heterogeneity, unlike a normal patient population [119]. A fully functional immune system is crucial in order to fully understand human response patterns and avoid therapeutic failure [118,120]. Molecular abnormalities of cancer originate from numerous parallel mutations affecting different pathways which cannot be recapitulated in laboratory mice at the same scale [119]. Human microbiome influences outcomes in some cancers, but the murine one is dramatically different and made even less relevant due to the impact of the artificial life habitat and feed. The murine model cannot accurately resemble the high complexity of spontaneously occurring tumours. A lot could be gained from introducing a new disease model closely resembling the human organism and immune response while avoiding limitations of the murine model [121].

### 7.2. Benefits of the Canine Model of Human Diseases

Numerous factors act in favour of comparative oncology research in dogs as a preclinical model for human disease. Cancer is the leading cause of fatality in dogs, affecting approximately one in four, and as many as 50% of dogs in certain predisposed breeds [122]. Despite a relatively long history of veterinary studies, there is an urgent need for novel therapeutics. Dogs constitute a unique model in that many canine cancers and their treatments are relevant to humans. At the same time, they age faster and develop spontaneous tumours, unlike the artificially induced tumour models in mice [123]. When the canine genome was sequenced in 2005 [124], it became clear that it is closer to the human one than the murine one is. Uniquely, companion dogs share the human owners’ lifestyle and risk factor exposures. Interestingly, there are many microbiome similarities between pet dogs and their owners [125]. Dogs possess a fully functional immune system and their population is highly heterogeneous. Investigating mechanisms underlying cancer in dogs and developing veterinary immunotherapeutics may be beneficial for both species.

### 7.3. Osteosarcoma Exemplifies Benefits of The Canine Model for Research into Human Immunotherapy and Cancer Progression

One of the most prominent diseases of unmet clinical need in both humans and dogs is osteosarcoma (OS), a fatal malignancy with poor prognosis in both species [126]. In the majority of cases, it is diagnosed with metastatic disease detectable in the lungs. Moreover, up to 80% of patients are believed to have micrometastasis [127,128]. One of the challenges in developing treatments for OS is its relatively low occurrence in humans. However, OS is 27 times more common in dogs [129]. Trials of pembrolizumab in osteosarcoma failed with only 1 in 19 patients responding to treatment [130]. In nearly 50% of cases, the disease burden increased by more than 50% compared to baseline, fitting many of the HP criteria [130]. The expression of PD-L1 by cancer cells is well established in both species, while expression of PD-1 was shown in human patients [70]. Consequently, we observed PD-1 expression in canine OS cell lines (unpublished data). Importantly, our preliminary findings suggest that PD-1 blockade may accelerate OS growth, and we previously showed the existence of putative tumour-intrinsic PD-1 signalling in the canine osteosarcoma [131]. Canine and human OS share many pathological, morphological, and genetic similarities. One of the most common shared mutations in the PI3K pathway is the loss of PTEN gene. Importantly, PTEN mutations are correlated with immunotherapy resistance [132,133]. Other genes frequently affected by mutations in the OS in both species include TP53, PI3K, MAPK, DMD and SETD2. Considering all the similarities, canines could serve as a superior model for research into the rapid progression and ICB resistance in osteosarcoma.

## 8. Conclusions

While the concept and existence of HP remain controversial and occasionally questioned, dialogue around it is increasing. Based on a PubMed search for “hyperprogression”, after the introductory publication by Champiat et al. in 2016 there were nine search hits the following year, then 27 in 2018, 45 in 2019 and just six in January 2020. The topic was also discussed in depth during a dedicated panel at the AACR 2019 meeting in Atlanta. The exploration of causative mechanisms behind HP and development of prediction/detection methods is urgent if we consider the increasingly complex landscape of registered trials applying multiple ICB together or in combination with other treatment approaches. Here we propose that factors such as tumour-intrinsic IC signalling, the impact of pH in the tumour microenvironment on mAb functionality, gut and tumour microbiome composition, patients race and endocrine status, ICs and FcRs polymorphisms and the impact of PD-1/PD-L1 blockade on the PD-1 interaction with its other ligand/s deserve attention in the process of elucidating HP mechanisms. We further suggest that the canine model of human cancer could naturally mimic the characteristics of human disease, including the heterogeneity of the patient population, and offer an advantage over the murine model. Increased understanding of hyperprogression will facilitate the development of methods to correctly predict personalised treatment responses, stratify patients in regard to the expected benefit and to detect an early need for therapy change. Targeted use of ICB could benefit all patients irrespectively of their response to immunotherapy. This would improve safety and efficiency profiles of current and future therapeutics. As pointed out by Houot and others [6,113], HP is as much of a challenge as it is an opportunity for the biotherapeutic field. Attentive investigation of the previously unknown HP mechanisms may enable completely new therapeutic approaches.

## Figures and Tables

**Figure 1 cancers-12-00804-f001:**
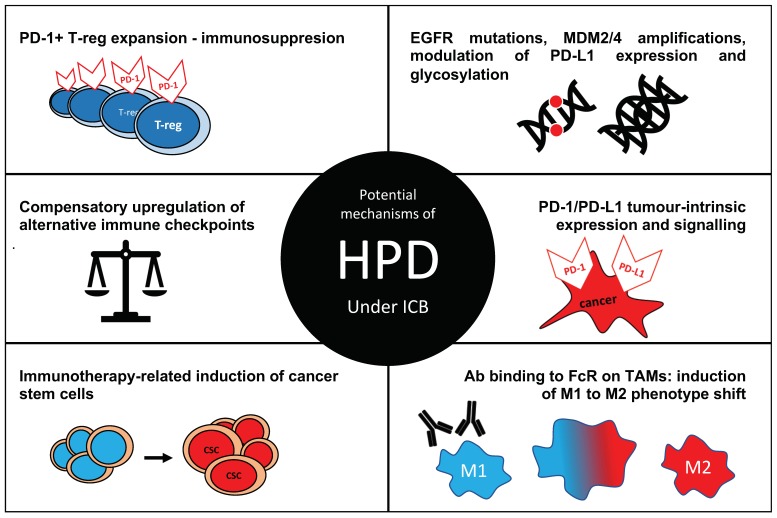
Postulated mechanisms of hyperprogressive disease (HPD) in cancer under immune checkpoint blockade (ICB). Ab—antibody, FcR—Fc receptor, TAM—tumour-associated macrophage; M1—a pro-inflammatory macrophage phenotype, M2—anti-inflammatory phenotype known to support tumour growth and metastasis.

**Table 1 cancers-12-00804-t001:** Therapeutic mAbs targeting human ICs.

Drug	Target	Type and Subclass	Indications	Manufacturer	Approval Status in EU and US
Pembrolizumab	PD-1	Humanised IgG4	Melanoma	Merck	Approved
Nivolumab		Human IgG4	Melanoma, non-small-cell lung cancer	Bristol-Myers Squibb	Approved
Dostarlimab		Humanised IgG4	Endometrial cancer	Tesaro (GSK)	Review
Cemiplimab		Human IgG4	Cutaneous squamous cell carcinoma	Sanofi	Approved
Toripalimab		Humanised IgG4	Unresectable or metastatic melanoma that failed previous systemic therapy [26]	Shanghai Junshi Bioscience	No/Conditional approval in China
Atezolizumab	PD-L1	Humanised IgG1	Bladder cancer	Genetech (Roche)	Approved
Avelumab		Human IgG1	Merkel cell carcinoma	Merck-Pfizer	Approved
Durvalumab		Human IgG1	Bladder cancer	AstraZeneca	Approved
Ipilimumab	CTLA-4	Human IgG1	Metastatic melanoma	Bristol-Myers Squibb	Approved
Tremelimumab		Human IgG2	Melanoma, mesothelioma, NSCLC	Medimmune (AstraZeneca)	Failed in trials
Balstilimab	PD-1^+^ CTLA-4	Human IgG4	Relapsed or refractory metastatic cervical cancer	Agenus Inc.	FDA Fast Track for the combination
with					
Zalifrelimab		Human IgG1			

Abbreviations: mAbs—monoclonal antibodies, ICs—immune checkpoints, PD-1—Programmed Cell Death Protein 1, PD-L1—Programmed Death-Ligand 1, CTLA-4—Cytotoxic T-cell Antigen 4, NSCLC—Non-Small Cell Lung Cancer, FDA—(United States) Food and Drug Administration.

**Table 2 cancers-12-00804-t002:** Criteria included in different HPD definitions.

Parameter Measured	[11]	[32]	[37]	[39]	[34]	[35]	[36]	[38]	[33] ^A^
Progression at the first evaluation (RECIST)	✓			✓		✓	✓	✓	
50% tumour burden increase		✓			✓				
TGRpost/TGRpre	≥2	≥2		≥1.5 ^B^		≥2	≥2	≥2	
TGKpost/TGKpre			≥2		≥2			≥2	
TTF < 2 months		✓					✓		3 of 5 ^A^
>50% increase in the sum of target lesions major diameters between the baseline and the first radiologic evaluation									3 of 5 ^A^
≥2 new lesions in an organ already involved at the first radiologic evaluation									3 of 5 ^A^
Spread to a new organ at the first radiologic evaluation									3 of 5 ^A^
ECOG decrease by ≥2 points in the first 2 months of treatment									3 of 5 ^A^

^A^ At least three of the proposed five criteria must be met; ^B^ Originally described as TGRpost−TGRpre > 50%; TGR—tumour growth rate according to RECIST criteria: TGRpre—the rate before treatment initiation, TGRpost—after treatment initiation; TGK(pre/post)—tumour growth kinetics—the current slope of the tumour growth; TTF—time to treatment failure; ECOG—Eastern Cooperative Oncology Group scale, describes the patient’s level of functioning in grades from 1 to 5.

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
