# Peer review of "Hyperprogression Under Immune Checkpoint-Based Immunotherapy—Current Understanding, The Role of PD-1/PD-L1 Tumour-Intrinsic Signalling, Future Directions and a Potential Large Animal Model"

_cancers, 2020, doi:10.3390/cancers12040804_

Round 1

Reviewer 1 Report

  1. The definitions of hyperprogressive disease differ by studies, and are controversial. It is helpful to summarize the currently suggested definitions in a table.
  2. It is also useful to summarize the previously reported studies to clarify the difference of the HPD according to cancer types, or the line of administration.
  3. Sometimes it is difficult to distinguish HPD from the situation that cancer is rapidly growing without relation to the administration of ICIs. In some cancers, ICIs are available for only salvage lines, and the response rate of conventional chemotherapy at that setting is also low.
  4. Recently, several clinical trials have combined ICIs with other ICI, cytotoxic agents or molecular target drugs. Are there any reports the frequency of HPD in those combination therapy?
  5. Why does the overexpression of TIM-3 on intratumoral T-cells occur only in HPD cases?
  6. Authors mentioned the PD-L1 expression is associated with HPD. Is it contradictory to the fact that PD-L1 expression level is considered as a predictive biomarker of the efficacy of ICIs in certain malignancies such as NSCLC?
  7. Authors referred the crossing of survival curves seen in the Checkmate-057 as the evidence of HPD. However, the early dropping of survival curve of ICIs does not necessarily represent the HPD.

Author Response

  1. Thank you for the comment. We have added a table, Table 2, with summary of current definitions proposed by multiple authors.
  2. Thank you for the comment. We have added the requested information in the text in lines 120-124. 
  3. Thank you for this insight. We agree with reviewers point of view, and this was discussed in 'Arguments agains HP' section starting in line 784
  4. Thank you for this comment. Unfortunately we did not find data pertaining to HP disease in combination therapies as a standalone publication. We have looked at several publications and unfortunately majority is lacking information on tumour growth rates/target lesion changes. We have included in the text one study, CheckMate-032 studying combination therapy of Nivolumab and Ipilimumab, the combination therapies (as two different dose schedules were used) actually resulted in increased number of patients with accelerated growth rates after starting the treatment (the data is presented in supplementary material).Text included in lines 306-316
  5. Thank you for the comment. The TIM3 over expression does not occur only in hyperprogressive cases, but overall in progressive/non-responsive tumours. We have modified this part of the text to clarify this statement (lines 301-304). 
  6. Thank you for the comment. We have clarified this section- in a single study an inverse correlation between PDL1 expression on cancer cells and HPD was identified (line 586).
  7. Again thank you for the comment. We have adjusted the sentence to incorporate reviewers comment.  

Reviewer 2 Report

The review by Kocikowski et al. summarizes and discusses available data on hyperprogression, a rapid disease progression occurring in cancer patients, particularly those receiving IC blockade.

The review is interesting, up to date and very well written.

Besides a few typing errors, I just have two criticisms

  • Line 59: PD-1 is also expressed by NK cells, as first demonstrated by the Caligiuri group (Blood. 2010. doi: 10.1182/blood 2010-02-271874)
  • The figure is poorly representative of what described in the review. My suggestion is to represent immune effector and target (tumor) cells, focusing on the expression/signaling in both cell types of immune checkpoint molecules. In addition, a figure/table summarizing possible mechanisms of hyperprogression could be appreciated.

Author Response

Thank you for your kind words and we are happy that you enjoyed reading our review. We have corrected the typos as requested. 

We have added NK cells in sentence describing cells expressing PD1 as requested.(line 61) 

We have modified the figures as requested- we moved the figure from the original manuscript to supplementary materials, we have also created a new figure which graphically shows currently hypothesised mechanisms of hyperprogression.

Round 2

Reviewer 2 Report

No further suggestions/comments